# Leveraging the Fragment Molecular Orbital Method to Explore the PLK1 Kinase Binding Site and Polo-Box Domain for Potent Small-Molecule Drug Design

**DOI:** 10.3390/ijms242115639

**Published:** 2023-10-27

**Authors:** Haiyan Jin, Jongwan Kim, Onju Lee, Hyein Kim, Kyoung Tai No

**Affiliations:** 1The Interdisciplinary Graduate Program in Integrative Biotechnology & Translational Medicine, Yonsei University, Incheon 21983, Republic of Korea; haiyanjin@yonsei.ac.kr (H.J.); lilyon@yonsei.ac.kr (O.L.); 2Department of Biotechnology, Yonsei University, Seoul 03722, Republic of Korea; 3Bioinformatics and Molecular Design Research Center (BMDRC), Incheon 21983, Republic of Korea; hikim@bmdrc.org; 4Baobab AiBIO Co., Ltd., Incheon 21983, Republic of Korea

**Keywords:** protein-protein interaction, fragment molecular orbital method, polo-like kinase 1, molecular dynamics simulation

## Abstract

Polo-like kinase 1 (PLK1) plays a pivotal role in cell division regulation and emerges as a promising therapeutic target for cancer treatment. Consequently, the development of small-molecule inhibitors targeting PLK1 has become a focal point in contemporary research. The adenosine triphosphate (ATP)-binding site and the polo-box domain in PLK1 present crucial interaction sites for these inhibitors, aiming to disrupt the protein’s function. However, designing potent and selective small-molecule inhibitors can be challenging, requiring a deep understanding of protein–ligand interaction mechanisms at these binding sites. In this context, our study leverages the fragment molecular orbital (FMO) method to explore these site-specific interactions in depth. Using the FMO approach, we used the FMO method to elucidate the molecular mechanisms of small-molecule drugs binding to these sites to design PLK1 inhibitors that are both potent and selective. Our investigation further entailed a comparative analysis of various PLK1 inhibitors, each characterized by distinct structural attributes, helping us gain a better understanding of the relationship between molecular structure and biological activity. The FMO method was particularly effective in identifying key binding features and predicting binding modes for small-molecule ligands. Our research also highlighted specific “hot spot” residues that played a critical role in the selective and robust binding of PLK1. These findings provide valuable insights that can be used to design new and effective PLK1 inhibitors, which can have significant implications for developing anticancer therapeutics.

## 1. Introduction

Polo-like kinases (PLKs) form a family of serine/tyrosine kinase proteins with wide distribution in eukaryotic cells and play crucial roles in various cell-cycle phases [1]. Currently, the PLK protein family comprises five members: PLK1, PLK2, PLK3, PLK4, and PLK5 (Figure 1A) [2]. Among these members, PLK1 has undergone the most comprehensive research to understand the regulatory mechanisms influencing its functions and potential as a target for drug design [3]. Generally, PLK1 exhibits a structure akin to other kinase family members, comprising an N-terminal serine/threonine kinase domain (KD) and a C-terminal repetition of the polo-box domain (PBD) [4]. The enzymatic activity of PLK1 is directly influenced by its phosphorylation [5]. Moreover, the expression pattern of PLK1 is linked to mitotic progression [6]. In particular, PLK1 is typically expressed at low levels during the interphase, and its expression gradually increases as the cell enters the G2/M phase, reaching its peak at this stage [7]. Following mitosis, PLK1 undergoes significant degradation, rapidly decreasing its protein levels [8]. During the cell cycle, PLK1 is crucial in regulating various processes. These include checkpoint recovery [9], the timing of mitotic entry [10], centrosome maturation, bipolar spindle assembly [11], microtubule-kinetochore attachment stabilization [12], and proper chromosomal segregation during anaphase [13].

Cell-cycle disruption is a prominent characteristic of various cancer types [14]. In malignant cells, there is an upregulation of PLK1 expression, leading to multiple defects in mitosis and cytokinesis. This heightened PLK1 activity increases chromosomal instability [15], often associated with a high tumor grade and unfavorable patient prognosis [14]. Beyond its role in cell-cycle dysregulation, PLK1 significantly promotes cancer progression via metabolic reprogramming, including enhancing the flux of the pentose phosphate pathway and directing glucose to pathways involved in the synthesis of macromolecules [16]. Furthermore, the downregulation of PLK1 expression typically reduces the proliferation of diverse cancer cells [17]. Thus, PLK1 is widely regarded as a potent proto-oncogene and a promising target for cancer therapy.

The PLK1 protein can be divided into two primary regions: the N-terminal kinase catalytic domain, which is highly conserved and spans approximately 252 amino acids to the 303 amino acids position, and the unique C-terminal PBD comprising two PBDs, each containing 60 to 70 amino acids [18]. The N-terminal catalytic KD (amino acid residues 53 to 303) possesses characteristic residues found in conserved serine/threonine kinases, including a T-loop and an ATP-binding cassette. This domain contains crucial residues, such as Lys82, Cys133, and Asp194, which are vital for ATP binding. Additionally, residues 134 to 136 form a functional bipartite nuclear localization signal sequence within this domain [19]. During the G2 phase of the cell cycle, the conserved residue 210 in the T-loop of PLK1 undergoes phosphorylation. This phosphorylation event is conducted by the upstream kinase Aurora A and its cofactor Bora, activating PLK1 [20]. The ATP-binding pocket of PLK1 consists of specific residues, namely Phe183 (at the bottom of the binding site), Leu59, Cys67, and Ala80 (at the top), thus effectively sandwiching the surface of the adenine ring system. The gatekeeper residues (Leu130 and Val129) are at the back of the adenine-binding pocket [21].

The C-terminal region of PLK1 (residues 345 to 603) primarily consists of two distinctive PBDs. The polocap located at the N-terminus of polo-box (PB) 1 can fold around PB 2, restricting its movement. Although PB 1 and 2 exhibit limited homology, their three-dimensional (3D) structures display remarkable similarities [18]. These proteins are pivotal in regulating PLK1 catalytic activity, localization, and substrate binding [22]. By binding to the PLK1 KD, these proteins act together to inhibit PLK1 kinase activity by inducing conformational changes in the protein. The interaction of the PLK1 PBD with Map205 stabilizes the autoinhibited state of PLK1 and sequesters it from its substrates [21]. Collectively, the flexible KD with its hinge region, along with the T-loop and the mentioned proteins, forms a switch that facilitates intricate spatiotemporal regulation of PLK1 activity [23]. The conformation of the PBD is crucial to its interaction with a phosphopeptide, enabling the main chains to interact and form a short antiparallel β-sheet between the peptide and PBD strand. Four specific residues, Trp414, Leu490, His538, and Lys540, significantly mediate other interactions. Particularly noteworthy are His538 and Lys540, which interact with the threonine phosphate group [24].

Comprising two domains, the KD and PBD, PLK1 offer inherent targets for PLK1 inhibitor development. Recently, researchers have made significant progress in creating numerous novel small-molecule PLK1 inhibitors, encompassing both KD and PBD inhibitors [25]. The ATP-competitive inhibitors target the deep groove in the kinase ATP-binding domain. At present, over 10 commercially available PLK1-specific inhibitors exist, with at least four of them (BI2536, BI6727 (Volasertib), GSK461364, and NMS-1286937 (Onvansertib)) having undergone evaluation in clinical trials (Table 1) [26]. All four PLK1 inhibitors function as ATP-competitive inhibitors and exhibit a similar mode of action. Among them, BI2536, a specific PLK1 inhibitor, has been extensively studied as a cytotoxic drug for treating various cancer types [27]. Nevertheless, BI2536 restricts antitumor activity during clinical trials and exhibits dose-limiting side effects [28]. In a recent study, the reduced effectiveness of BI2536 against progressive hepatocellular carcinoma was linked to low intratumoral drug levels [29]. Also known as Volasertib, BI6727 has exhibited great promise as a PLK1 inhibitor. Several preclinical experiments have demonstrated its effectiveness in inducing tumor regression [30]. Onvansertib, an orally administered and remarkably potent PLK1 inhibitor, exhibits high specificity for PLK1, excellent bioavailability, and a short half-life. Notably, it can potentially mitigate the toxicity of previous nonspecific PLK1 inhibitors [31].

Despite the promise of ATP-competitive inhibitors, resistance often emerges in cancer patients due to the high conservation of ATP-binding domains among various kinases and frequent mutations in ATP-binding sites [32]. Furthermore, ATP-competitive inhibitors may exhibit activity against other kinases, leading to a lack of specificity for PLK1.

In contrast, the PBD is unique to PLKs, making it a more viable and specific target for developing selective PLK1 inhibitors. Three selective PLK1 inhibitors (poloxin, thymoquinone, and purpurogallin) target the PBD of PLK1 [33]. Both poloxin and thymoquinone can obstruct the correct orientation of PLK1, effectively impeding the mitosis of cancer cells [34]. A recent study revealed that poloxin-2, an optimized analog of poloxin, exhibits substantially enhanced potency and selectivity compared to poloxin. This improvement efficiently induces mitotic arrest and apoptosis in cultured human cancer cells [35]. As mentioned previously, PLK1 inhibitors have dose-limiting toxicities. Therefore, various approaches, such as bivalent inhibitors, were introduced. Andrej and his colleagues successfully identified the initial bifunctional inhibitors of PLK1, which are bridge kinase inhibitors and PBD peptides [36].

In 1999, Kitaura et al. [37] introduced the Fragment Molecular Orbital (FMO) method, partitioning a target molecule into smaller fragments for molecular orbital calculations [37,38]. The FMO method has recently gained popularity in new drug designs due to its accurate energy calculations, particularly in analyzing protein–ligand and protein–protein interactions (PPIs) [39,40]. By employing the FMO method, we can calculate fragment–fragment interaction energies, providing valuable insight into protein–ligand interactions. The energies of fragment–fragment pairs are called inter-fragment interaction energy. Nakano et al. developed packages in ABINIT-MP that implement the FMO method [41]. The software suite incorporates pair interaction energy (PIE) decomposition analysis (PIEDA) to estimate inter-fragmental interactions between functional group units based on their distinct contributions from electrostatic and dispersion forces.

Recently, the FMO method has gained recognition as a powerful tool for analyzing biological interactions between ligands and proteins in various contexts, such as estrogen receptors [42], human immunodeficiency virus proteases [43], influenza neuraminidases [44], G protein-coupled receptors, serine/threonine protein kinase Pim-1 [45], and YAP binding transcription factor TEAD [46]. This study involved an FMO/3D scattered PIE (SPIE) analysis to explore interaction systems at the ATP-binding site. Additionally, to investigate PPIs, the FMO/3D-SPIE analysis was introduced to identify crucial interactions in previous work. This analysis efficiently correlated the results with experimental site-directed mutagenesis findings [47].

In this study, we employed the Fragment Molecular Orbital (FMO) and 3D-SPIE analyses to investigate protein–protein interactions (PPIs) between the substrate peptide and the polo-box domain (PBD) of polo-like kinase 1 (PLK1) at the quantum mechanical level. We utilized the FMO method for an in-depth analysis of interactions within the ATP-binding site and the PBDs, which was integral to our hot-spot analysis of PLK1. This research delves into the ATP and substrate peptide binding sites of the kinase domain (KD) and PBD of PLK1, respectively. We examined the interactions with inhibitors BI2536, Onvansertib, and GSK461364, illuminating the key residues responsible for selectivity. Our FMO analysis provided insights into their distinct binding modes, emphasizing the significant role of specific PBD binding pockets. We explored PBD’s binding to peptides and small molecules, pinpointing the “hot spot” regions crucial for potency. Molecular dynamics (MD) simulations and solvation analysis further validated our results, highlighting potential strategies for augmenting inhibitor efficacy. Collectively, this research offers invaluable insights into drug design, enhancing our understanding of PLK1’s binding mechanisms.

## 2. Result

### 2.1. Hot-Spot Analysis for the Kinase Domain in PLK1

In this first stage of this work, we investigated hot spots of PLK1 using FMO analysis. We focused on two major druggable binding regions of PLK1: the kinase- and PBD-binding sites (Figure 1B). The complexes in each domain were derived from the protein data bank (PDB). To investigate the KD in PLK1, we compared the binding pose for known inhibitors (Figure 2A). Two classes of binding pose tendency were observed among the inhibitors. The A group was expanded to the solvent exposure site with the para position of the benzene group, whereas the B group was expanded to the solvent exposure site with the meta position of the benzene group, which can directly interact with Glu140. The conserved core moieties among the inhibitors are illustrated in Figure 2B. Thus, we used three crystal structures of the most potent inhibitors, including ATP (Figure 3A, PDB ID: 2OU7), BI2536 (Figure 3B, left), and Onvansertib binding structures (Figure 3B, right, PDB ID: 2YAC).

We also analyzed the binding structures of known compounds, which are not revealed as the crystal structure, GSK461364. We performed molecular docking and MD simulations to predict the noncrystal structure binding pose. We considered any interaction with an absolute ΔE^int^ ≥ 3.0 kcal/mol significant and treated the residues involved in this interaction as hot spots. In addition, ΔE^int^ is represented as a PIE plot, and decompositions of intermolecular interactions for ΔPIE, which is divided into five energy terms: electrostatic (∆E^es^), exchange-repulsion (∆E^ex^), charge transfer with a higher-order mixed term (∆E^ct+mix^), dispersion (∆E^di^), and solvation energy (∆G^sol^). Furthermore, we implemented an advanced analysis technique involving an MD simulation and solvation analysis to deepen our understanding of the binding mode of KD inhibitors. This approach was pursued while considering the dynamic behavior of the protein–ligand complex and the concomitant movement of water molecules. This advanced methodological approach offers a comprehensive perspective on the interplay between the inhibitor and the KD, underlining the significance of water molecules in influencing this interaction.

#### 2.1.1. Fragment Molecular Orbital Analysis for Kinase Domain: ATP-Binding Site

To investigate the KD of PLK1, first, we analyzed the ATP-binding structure using the FMO method. The crystal structure of the human PLK1 KD with ATP (PDB ID: 2OU7) was subjected to an FMO analysis. The ATP-binding site is classified into the adenine, ribose, and phosphate pockets and solvent channels (Figure 3A) [48]. The FMO result of the holo ATP-binding structure is presented in Figure 4 and Appendix A. In the adenine pocket, the adenine group interacts with the backbone of the Cys133, which has −10.425 kcal/mol PIE (Appendix A). Moreover, Phe183 can interact with aromatic rings from adenine moiety and has significant binding energy. Although there is no direct interaction during visual inspection in the ribose pocket, Gly62, Arg136, and Cys67 contribute essential energy to the ribose part. Lys82 has the lowest PIE in the phosphate pocket with ATP, which binds with the phosphate part. Many groups have already demonstrated that when Lys82 mutates to Met, the kinase activity loses function, and there is no effect on S-phase progression [49]. These results suggest specific interactions between the ATP and KD of PLK1, with specific amino acids playing critical roles in stabilizing the binding.

#### 2.1.2. Fragment Molecular Orbital Analysis for Kinase Domain: BI2536 and Onvansertib

We investigated the X-ray crystal structures of known inhibitors with nanomolar potency: BI2536 has an efficacy of IC_50_ 0.83 nM, and Onvansertib has an efficacy of IC_50_ 2 nM (PDB ID: 2RKU, 2YAC). Although both compounds exhibit a similar nanomolar range potency, only Onvansertib demonstrates selectivity for PLK1 over PLK2 and PLK3 [50]. Additionally, BI2536 and Onvansertib display markedly different binding modes, particularly in the solvent channel region (Figure 3B). Additionally, BI2536 occupies the solvent Channel 1 position with N-methylpiperidine, whereas Onvansertib occupies the solvent Channel 2 position with N-methylpiperazine (Figure 3B). The results of the FMO analyses of BI2536 and Onvansertib with PLK1 are depicted in Figure 5A,B. Further, BI2536 exhibits significant interaction energy with 10 amino acids of PLK1: Arg57, Leu59, Gly60, Gly62, Cys67, Glu69, Leu132, Cys133, Arg136, and Phe183 (Appendix A). Onvansertib also displays significant interaction energy with 14 amino acids in PLK1: Arg57, Leu59, Gly60, Cys67, Lys82, Leu132, Cys133, Arg134, Arg136, Ser137, Glu140, Asn181, Phe183, and Asp194 (Appendix A). These two inhibitors share common core interactions with PLK1 involving Arg57, Leu59, Gly60, Cys67, Leu132, Cys133, Arg136, and Phe183. Particularly, Cys133 is a key residue for hinge-binding interactions in the backbone atoms forming two hydrogen bonds with the inhibitors. Onvansertib features unique interactions, distinguishable from BI2536, through the N-methylpiperazine moiety with robust electrostatic interactions with Glu140 in PLK1 and the amide moiety, forming strong hydrogen bonds with Lys82 and Asp194 in PLK1.

#### 2.1.3. Fragment Molecular Orbital Analysis for Kinase Domain: GSK461364

Like Onvansertib, GSK461364 demonstrates the same hinge interactions in the benzimidazole moiety with Cys133 in PLK1, significantly strong electrostatic interactions in the N-methylpiperazine moiety with Glu140 in PLK1, and strong hydrogen bonds in the amide moiety with Lys82 and Asp194 (Figure 6A). Although the drug structures of GSK and Onvansertib differ, the molecular binding tendency toward PLK1, via the FMO analysis, was consistent in both cases (Figure 6 and Appendix A). Onvansertib and the GSK461364 drug possess selectivity toward PLK1, among other PLK series [51]. Compared to drugs that lack selectivity for PLK1, such as BI2536, these two drugs have been confirmed to bind to Glu140 of PLK1 specifically. Similar to Onvansertib, GSK461364 demonstrates the same hinge interactions with Cys133 in PLK1 via the benzimidazole moiety, along with significantly strong electrostatic interactions with Glu140 in PLK1 via the N-methylpiperazine moiety. A notable hydrogen bond interaction with Lys82 and Asp194 in PLK1 via the amide moiety also exists (Figure 6), implying that, despite the distinct structural differences between the GSK and Onvansertib drugs, their molecular binding tendencies toward PLK1 are consistent, as deduced from the FMO analysis. Onvansertib and GSK461364 exhibit selectivity toward PLK1 over other PLK series [51]. The superposition of the crystal structures of PLK1–3 and the sequence alignment are presented in Figure 7. The N-methylpiperazine moiety might play a vital role in imparting selectivity to the compound toward PLK1 in contrast to PLK2–3, as it establishes a polar interaction with the side chain of Glu140. Nevertheless, in PLK2 and PLK3, where Glu140 is substituted with histidine, the formation of the polar interaction is impeded (Figure 7).

#### 2.1.4. Molecular Dynamic Simulation and Solvation Analysis for the Kinase Domain

We conducted a comprehensive MD simulation analysis of the holo-form of the KD by performing 12 crystal structure determinations (Appendix A) and a docking structure of GSK461364, each for a duration of 100 ns. This approach ensured that the initial input docking structure did not bias the results. Subsequently, we analyzed the trajectories of each complex from the simulations by assessing ligand–protein contacts. The percentage in the findings represents the frequency of interactions occurring during the simulation. We depicted the summary of these results in a heatmap, as illustrated in Figure 8. Remarkably, all 13 ligands displayed a consistently high percentage of interaction with the hinge residue Cys133. In the phosphate pocket (Figure 3A), ligands from Group A interacted with Lys82 and Asp194 via a water bridge. In contrast, Group B exhibited direct and robust interactions with the residues in the phosphate pocket. Only ligands from Group B demonstrated an interaction with Glu140, a selective residue among PLK1, PLK2, and PLK 3. Based on the patterns observed in the heatmap, GSK461364 likely belongs to Group B.

Using WaterMap calculations, we performed a solvation analysis that effectively identified specific hydration sites localized near the ATP-binding site, providing valuable insight into their thermodynamic energy profiles [52]. Hydration sites with a predicted free energy (ΔG) greater than 0 kcal/mol upon water addition were classified as unfavorable sites, whereas hydration sites with a predicted free energy ΔG of less than 0 kcal/mol were considered favorable sites [53]. A comprehensive thermodynamic analysis of the hydration sites has significant importance in drug design because removing high-energy hydration sites from the protein binding site is a critical factor influencing binding affinity [54]. Comparing the WaterMap results for BI2536 and Onvansertib, we observed an unfavorable hydration site at the carbonyl position of Onvansertib, forming a hydrogen bond with Lys82 in the phosphate pocket (Figure 9A), which was absent in the WaterMap analysis of Onvansertib (Figure 9C). Furthermore, two additional unfavorable hydration sites were localized at the N-methylpiperazine position, which did not coincide with the corresponding positions in the WaterMap of Onvansertib (Figure 9B,D). These results suggest that Onvansertib obtains selectivity by functional groups by replacing high-energy hydration sites found in the WaterMap of BI2536.

### 2.2. Hot-Spot Analysis for the Polo-Box Domain in PLK1

In the subsequent phase of the investigation, attention was primarily focused on the PBD of PLK1. Via the application of FMO calculations, we scrutinized its PPIs with the substrate peptide of polo-box interaction protein 1 (PBIP1), two peptide mimic inhibitors (4j, 4a), and a small molecule (KBJK557). Primarily, we evaluated the PPI hot-spot region of PBD and the substrate peptide of PBIP1 in depth. Subsequently, we performed the FMO analysis on the most potent peptide ligands and small-molecule inhibitors. Furthermore, we also performed the MD simulation and solvation analysis to gain insight into the binding mode of PBD ligands, considering the movement of the protein–ligand complex and water molecules. As a result of the comprehensive investigation, we acquired crucial insight suggesting that enhancing inhibitor potency could be effectively achieved by focusing on specific key points discovered in this study.

#### 2.2.1. Fragment Molecular Orbital Analysis for the Polo-Box Domain: PBIP1-PBD

As mentioned, the PBD substrate binding pocket can be divided into phosphate, pyrrolidine, and Tyr-rich pockets [55] (Figure 10). The crystal structure (PDB ID: 3P37) of the PBD complex with FDPPLHSpTA phosphopeptide from PBIP1 [56] was calculated using the FMO method to investigate the hot spot of PBIP1-PBD interactions. The FMO results of the substrate peptide interaction with the PBD are presented in graphs (Figure 11). The substrate peptide was divided into nine fragments: Ala1, p-Thr2, Ser3, His4, Leu5, Pro6, Pro7, and Phe9 (Figure 11A). These nine fragments have noticeable contact with the PBD at 12 residues, with PIE values lower than −3 kcal/mol. The p-Thr2 primarily interacts with Lys540 and His538 at −71.283 kcal/mol and −67.246 kcal/mol, respectively. The backbone of Ser3 interacts with the backbone of Trp414 from the PBD via two hydrogen bonds (Figure 11B). These results agree with previous studies that the immunoprecipitation of wild-type PBIP1 strongly interacts with the PBD, but the active signal disappears with one of the serine77 (Ser3) and threonine78 (p-Thr2) residues [57]. Further, Ala1, p-Thr2, and Ser3 occupy phosphate and pyrrolidine pockets, which maintain high energy with Leu490, Leu491, His538, Lys540, and Trp414 (Figure 11B, right table). The remaining seven residues contribute interaction energy to the Tyr-rich pocket, displayed as a light-pink surface (Figure 10) and a light-pink stick (Figure 11B). In particular, Tyr417 from the Tyr-rich pocket interacts simultaneously with Leu5, Pro6, Pro7, Asp8, and Phe9 from the substrate peptide. Sharma et al. demonstrated that Tyr-rich pocket mutation selectivity impairs PBD binding to the substrate protein PBIP1 [58]. The mutagenesis data are related to the FMO results in this study.

#### 2.2.2. Fragment Molecular Orbital Analysis for the Polo-Box Domain: Peptide Ligands

Ligand 4j is the most potent peptide mimetic ligand developed from the PLHSpT sequence and has an alkyl-phenyl group with a PBD-binding IC_50_ value of 0.12 μM. However, the PBD binding with a 4a IC_50_ value is more than 200 μM, as the alkyl –phenyl group is absent [59]. We calculated PIE values with the crystal structure of the 4j binding in the PBD (PDB ID: 3RQ7) and the docking model 4a in the PBD using the FMO method. Two peptide ligands were fragmented based on peptide bonds (Figure 12A and Figure 13A). In the phosphate and pyrrolidine pocket, the p-Thr2 and Ser3 fragments from 4j have more contribution energy residues than 4a (Figure 12 and Figure 13). In particular, the PIE between 4j with His538 and Lys540 is much lower than 4a (Figure 12 and Figure 13). We found more significant energy differences between the two peptide ligands in the Tyr-rich pocket. Seven essential key residues have significant PIE values with F-Akyl from 4j, but only four with F-CH3 from 4a (Appendix A). Moreover, Tyr417 has the most frequent interactions with substrate peptides and has significant interaction energy with 4j but not with 4a. Thus, the FMO results of the two peptides are related to those from the structure analysis relationship mentioned above [59].

#### 2.2.3. Fragment Molecular Orbital Analysis for the Polo-Box Domain: Small Molecules

In addition, KBJK557 was identified as a potent PBD-binding small molecule, confirming that a four-carbon alkyl chain attached to pyrazole nitrogen might increase the potency via structure–activity relationship (SAR) studies [60]. For negative control, we also chose KBJK-4a, which does not have the four-carbon alkyl chain, to perform the in silico analysis. First, we performed the docking generation of KBJK557 in the PBD to determine the best pose with the lowest Glide Emodel score suitable for pose selection. In this best pose, the barbituric acid interaction with Lys540, His538, and Trp414 formed hydrogen bonds and a salt bridge, and the pyrazole part exhibited π-π stacking interaction with Trp414 (Appendix A). Furthermore, the phenyl group in the four-carbon alkyl tail engaged in a π-π stacking interaction with Tyr417 (Appendix A). Second, we ran the 50ns MD simulation of the docking structure to optimize it. We chose the optimized structure from Frame 489 of the MD simulation trajectory for the next step (Appendix A and Figure 14A). From the analysis of the MD simulation, the interactions found with the docking structure continued during the simulation (Appendix A). We also docked the KBJK-4a in the protein structure from Frame 489, and those interactions in KBJK557 almost exist except for the π-π stacking interaction with particularly Tyr417 (Figure 14D).

Finally, we performed the FMO study for two small-molecule complex structures (Figure 14). The KBJK557 interaction residues in the PBD were His538, Lys540, Trp414, Asp416, Tyr485, Tyr417, Leu490, Val415, Leu491, and Phe535 (Figure 14A and Appendix A). Some of these residues have been mentioned in previous hot-spot analyses for PBIP1-PBD PPIs and peptide ligands. Moreover, His538 and Lys540 interacted most with KBJK557, with PIE values of −25 kcal/mol and −23.556 kcal/mol. Electrostatic terms primarily drove these interactions. The Trp414 contacted KBJK557 with a PIE of −21.232 kcal/mol, driven by electrostatic and dispersion terms. The KBJK557 also engaged in hydrophobic interactions in the Tyr-rich pocket with Val415, Tyr417, and Tyr485 (Figure 14B,C). However, the interaction with Tyr417 disappeared in KBJK-4a (Figure 14E,F and Appendix A). These results are related to the previously mentioned analysis of peptide ligands 4j and 4a (Figure 12, Figure 13 and Appendix A). Moreover, these results align with previous SAR studies [60].

To gain insight into increasing the inhibition potency of the PBD in PLK1, we summarized all FMO results into one heatmap (Figure 15). We listed the amino acids of the PBD selected within 5 Å of the ligand and highlighted the critical residues in each pocket. The PIE was summed for each residue, especially for the substrate peptide and peptide mimetic ligands. Except for KBJK-4a, no dramatic energy changes occurred between the interaction with His538, Lys540, and Trp414 from the phosphate and pyrrolidine pockets (Figure 15). However, in the Tyr-rich pocket, peptide ligand 4a, and small molecule KBJK-4a, which have no or low potency in the experiment [58], the PIE became unstable with Tyr417, Tyr481, Phe482, and Tyr485 (Figure 15). The PIE of KBJK557 with Tyr481 and Phe482 was also reduced because the phenyl group in the four-carbon alkyl chain was short of occupying the entire Tyr-rich pocket (Appendix A). Taken together, the heatmap of the FMO results of five structures illustrates that the Tyr-rich pocket, in particular, Tyr417, Tyr481, Phe482, and Tyr485, may include crucial residues to contribute to the binding affinity of the PBD. These results relate to the mutagenesis experiments from the Sharma groups mentioned above [58].

#### 2.2.4. Molecular Dynamics Simulation and Solvation Analysis for the Polo-Box Domain

We assessed complex structures, including substrate peptide PBIP1, two peptide ligands, and KBJK557 of the PBD, for 100 ns. In KBJK557, we used the selected snapshot structure of KBJK557 as an initial structure for MD simulation. Then, the trajectory of each complex for each run was analyzed using ligand–protein contacts. The percentage represents interactions that occur during the simulation time. The heatmap of the protein–ligand contact from the MD simulation is presented in Figure 16, and ligands represent substrate peptides, peptide ligands, and small molecules that bind in the PBD. The Trp414 from the pyrrolidine pocket maintains a high percentage with all ligands, which were also displaced in the FMO analysis (Figure 15 and Figure 16). This outcome indicates that Trp414 is an essential residue in the PBD-binding site, which is in the central position. Peptide inhibitor 4j has a contact map similar to the substrate peptide, but 4a loses the interaction contact in the Tyr-rich pocket. Small molecules, KBJK557 and KBJK-4a, demonstrate mediate interaction contact with His538 and Lys540 compared to the substrate and peptide ligands—residues belonging to the phosphate-binding pocket (Figure 16)—which might be why the potency of small molecules is lower than that of the peptide ligands.

In the solvation analysis for the PBD, we might discover why the potency is different between 4j and 4a. In the WaterMap of 4a, three unstable hydration sites occupy the al-kyl-phenyl moiety position of 4j, forming a π-π stacking interaction with Phe482 and Tyr417 (Figure 17A,B). However, these hydration sites are not exhibited in the WaterMap of 4j (Figure 17C). Four unfavorable solvation sites were also found at the same position in the WaterMap of KBJK557, around the Tyr-rich pocket (Figure 17D). These results suggest that efforts to improve the inhibition potency of the PBD should focus on extending the moieties into the Tyr-rich pocket.

## 3. Discussion

A group of PLKs, consisting of five serine/threonine kinases, can be found in different eukaryotic organisms [61]. These kinases play a crucial role in regulating cell proliferation, particularly in controlling the progression of the cell cycle [4]. The PLK1 protein consists of an N-terminal serine/threonine kinase domain (KD) and a C-terminal repeat of the polo-box domain (PBD), with the phosphorylation of the latter directly affecting the enzymatic activity of PLK1 [55]. The PBD recognizes phosphorylated serine/threonine protein substrates to regulate PLK KD’s phosphorylation activity [22]. Each domain comprises druggable binding sites, including an ATP-binding site in the KD and a substrate peptide binding site in the PBD. Our computational study investigates two druggable binding sites to improve selectivity in the KD and potency in the PBD.

Numerous ATP-competitive inhibitors have been developed, and some have even advanced to clinical trials (as indicated in Table 1). However, these ATP-competitive inhibitors have shown limitations in antitumor activity during clinical trials, often exhibiting dose-dependent side effects. Consequently, a recent focus has been on inhibitors that bind to the PBD instead [48]. In this study, we focus on analyzing the ATP binding site and investigating the binding site in PBD to design an improved inhibitor. We analyzed using the FMO method, MD simulations, and solvation analysis.

In the former part of our study, we analyzed the ATP-binding site in the kinase domain. It is widely recognized that the adenine group of ATP interacts with the backbone of the hinge residue in kinase inhibitors [62]. Our FMO study shows that PLK1 inhibitors strongly interact with Cys133 via their core scaffold, as reflected in the high absolute PIE value. These findings underscore the pivotal role of hinge interactions in achieving the high potency of ATP-competitive inhibitors. However, the inhibitors of PLK1 that are ATP-competitive have exhibited a consistent problem with selectivity towards PLK2 and PLK3. To address this challenge, we have highlighted the significance of Glu140 in PLK1. In contrast, the corresponding positions in PLK2 and PLK3 are His169 and His149, respectively (Figure 7). This insight arose from the analysis of two well-known inhibitors, BI2536 and Onvansertib, both of which exhibited nanomolar potency against PLK1. Notably, only Onvansertib demonstrated selectivity for PLK1 over PLK2 and PLK3. While both inhibitors share common core interactions, Onvansertib exhibits additional interactions facilitated by specific moieties, such as N-methylpiperidine, contributing to its unique selectivity and binding properties. The N-methylpiperazine moiety in Onvansertib plays a crucial role in conferring selectivity towards PLK1, primarily due to its polar interaction with the side chain of Glu140. This insight deepens our understanding of the SAR of these inhibitors and offers guidance for the development of more potent and selective ATP-competitive PLK1 inhibitors in the future.

In the latter part of our study, we delved into identifying the binding site for the substrate peptide within the PBD. Over the years, a range of ATP-competitive inhibitors have been crafted, with several advancing to clinical trial stages. However, their effectiveness has often been overshadowed by dose-dependent adverse reactions. As a result, there has been a growing interest in inhibitors targeting the PBD. Yet, the design of nanomolar small molecule inhibitors specific to the PBD remains a formidable task. To navigate this, we probed the PBD substrate binding pocket, aiming to elucidate strategies to enhance inhibitor binding potency. Our FMO analyses reasonably pinpointed the Tyr-rich pocket as an essential component in enhancing PBD inhibitor efficacy. In our quest, we evaluated both peptide ligands and small molecules for which SAR data were accessible. Our focus was on contrasting ligands equipped with a binding moiety for the Tyr-rich pocket against those devoid of it. Notably, ligands lacking this specific moiety yielded considerably diminished absolute PIE values in FMO calculations or manifested heightened energy solvation sites. This underscores the potential of tailoring ligand moieties to fit the Tyr-rich pocket better, positing it as a potent strategy to optimize PBD inhibition. Our discoveries offer profound insights, enriching our understanding of binding dynamics and paving the way to design more potent PBD inhibitors.

Collectively, our results compellingly advocate for targeting both KD and PBD pockets with bivalent inhibitors as a potent approach to inhibit PLK1. Bivalent kinase inhibitors, characterized as a novel subset of small molecule compounds, are designed to engage two binding sites on kinase enzymes concurrently. These inhibitors stand out, offering a distinct edge over traditional inhibitors by simultaneously targeting both the ATP-binding (active) site and another site, typically termed as an “allosteric” or “regulatory” site [63,64]. Recent studies from other research groups have showcased the efficacy of bivalent inhibitors that bridge KD and PBD inhibitors for PLK1 [36,65]. Such groundbreaking strategies, epitomized by bivalent inhibitors, hold the potential to overcome the clinical limitations associated with ATP-competitive inhibitors of PLK1. The conception and development of these bivalent inhibitors demand a synergistic fusion of structural understanding, advanced computational modeling, and nuanced medicinal chemistry. Anchored by the findings of our study, we believe we stand ready to provide indispensable insights for the next wave of bivalent PLK1 inhibitor designs.

## 4. Materials and Methods

### 4.1. Protein Structure Preparation

Fourteen crystal structures were retrieved from the PDB (Appendix A). The structure was prepared using the Protein Preparation Wizard tool. All missing loops were filled using Prime implemented in the Maestro program (v. 2022; Schrödinger, LLC, New York, NY, USA, 2022). Hydrogen atoms were added to the complex structure at a pH of 7.4, and their positions were optimized using PROPKA implemented in the Maestro program. Restrained energy minimization was performed with an OPLS4 force field within a 0.3 Å root mean square deviation.

### 4.2. Molecular Docking

Molecular docking was conducted to generate the docking pose of GSK461364, KBJK557, and KBJK-4a using Glide [66] (Schrödinger, LLC, New York, NY, USA, 2022). In addition, GSK461364 was docked in the KD of PLK1, and KBJK557 and KBJK-4a were docked in the PBD of PLK1.

### 4.3. Fragment Molecular Orbital Calculations

All FMO computations were computed using the GAMESS 22 program [67]. The two-body FMO method was employed in all such calculations, with each residue in the protein and ligand being classified as a fragment. Per the hybrid orbital projection scheme fragmentation [68], an in-house code was used to prepare all input files.

The two-body FMO method comprises four stages: (1) fragmentation, (2) single-fragment self-consistent field (SCF) calculation, (3) two-fragment SCF computation, and (4) total property evaluation, the specifics of which have been previously detailed [69]. During the fragmentation phase (1), each residue in the protein, ligand, and water molecules can be designated as a fragment. Unlike regular peptide bond separation in proteins, all residues were divided at the sp3 bond between the alpha carbon and carbonyl carbon atoms in the backbone structure per the hybrid orbital projection scheme [68]. This method significantly decreases computational expenses and corrects inaccuracies from artificial fragmentation using a projection operator [68].

In the second and third stages (2, 3), all molecular orbitals (MOs) on a fragment were optimized using the SCF theory in the overall electrostatic field, and all electron densities were self-consistently resolved using self-consistent charge iterations [38,69]. The difference between these stages lies in the Hamiltonian operators [38]. The second stage optimized all MOs in a fragment, including the electrostatic potential from N-1 fragments, whereas the third stage optimized all MOs in two fragments and included the potential from N-2 fragments.

All MO results obtained in Stages 2 and 3 were combined to assess the system’s total properties (4) [69]. The PIE values between two fragments were computed, and the energy decomposition of the PIE values was conducted to comprehend the contributions of the five energy terms. These PIE values between fragments in the FMO computations were divided into five energy terms, as defined in Equation (1): electrostatic (∆E^es^), exchange-repulsion (∆E^ex^), charge transfer with a higher order mixed term (∆E^ct+mix^), dispersion (∆E^di^), and solvation energy (∆G^sol^) derived from the polarizable continuum model (PCM):∆E^int^ = ∆E^es^ + ∆E^ex^ + ∆E^ct+mix^ + ∆E^di^ + ∆G^sol^(1)

To scrutinize the crucial interactions of several protein–peptide complex or protein–ligand structures, we applied the second-order Møller–Plesset perturbation theory (MP2) [70] and PCM [71,72] with a 6–31G** basis set (FMO2-MP2/6–31G**/PCM level) to the protein–ligand or protein–peptide complex. We also calculated the crystal structure of the PLK1 complex with ATP using the FMO2/DFTB3 method [73]. Subsequently, we carried out significant PPIs in the complex by selecting PIE values that were more stable than −3.0 kcal/mol and had a distance of less than 4.0 Å between two fragments [45].

### 4.4. Molecular Dynamics Simulation

The MD simulation was performed using Desmond in Maestro (v. 2022-4; Schrödinger, LLC, New York, NY, USA, 2022). The 14 protein–ligand crystal complexes (Appendix A) and docking structure of GSK461364, 4a, KBJK557, and KBJK-4a were inserted into an orthorhombic box filled with explicit water molecules (TIP3P model) and with a buffer distance of 10 Å. The MD simulation was studied, and the OPLS4 force field was used. Ions (Na^+^ and Cl^−^) were added to simulate a physiological concentration of monovalent ions (0.15 M). A constant number of particles at a constant temperature of 300 K with a pressure of 1.01325 bar was used as an ensemble class. The particle-mesh Ewald method [74] was used to calculate long-range electrostatic interactions, with a cutoff for van der Waals and short-range electrostatic interactions of 9 Å. Nose–Hoover thermostats [75] were employed to maintain a constant simulation temperature, whereas the Martina–Tobias–Klein [76] method controlled the pressure. A RESPA integrator [77] integrated the equation of motion with an inner time step of 2.0 fs for bonded and non-bonded interactions within the short-range cut-off. The default protocol in Desmond was applied to reach system equilibration, and a 100 ns simulation was performed for each complex [78]. The analysis plots and Figures were sketched using the Desmond simulation interaction diagram panel of the Maestro program.

### 4.5. WaterMap Calculations

WaterMap, which has been extensively explained in other sources, employs a molecular dynamics methodology to forecast the thermodynamic characteristics of water molecules within a protein setting, encompassing entropy, enthalpy, and free energy for each water site [79]. The calculations for WaterMap (v. 2018-4; Schrödinger, LLC, New York, NY, USA, 2018) were conducted using default parameters, involving a 2 ns simulation time, with water molecules analyzed within a 10 Å radius of the ligand and using the prepared cocrystal structures (Appendix A) and docking structures mentioned in this paper with each ligand present.

## 5. Conclusions

In conclusion, our study has integrated FMO methodology, MD simulations, and solvation analyses to delve into the KD and PBD domains of the PLK1 protein. Our findings obtained via computation are in agreement with the results of biological assays conducted by other research groups. These insights illuminate the complex interactions and binding preferences among various PLK1 inhibitors. Importantly, our research highlights the consistent binding patterns of ATP-competitive inhibitors to PLK1, underscoring their selectivity. Additionally, our exploration of the PBD’s substrate peptide binding site identified crucial residues essential for interactions, suggesting pathways to enhance inhibitor efficacy. Collectively, these discoveries significantly enhance our understanding of the mechanisms governing binding, selectivity, and potential therapeutic pathways for PLK1-targeting inhibitors. Drawing from our research, we can provide valuable guidance for designing future bivalent PLK1 inhibitors with meaningful clinical implications. This wealth of information will undoubtedly shape the development strategies of forthcoming pharmaceuticals, guiding the creation of more effective and selective inhibitors that control cell growth and direct the cell cycle.

## Figures and Tables

**Figure 1 ijms-24-15639-f001:**
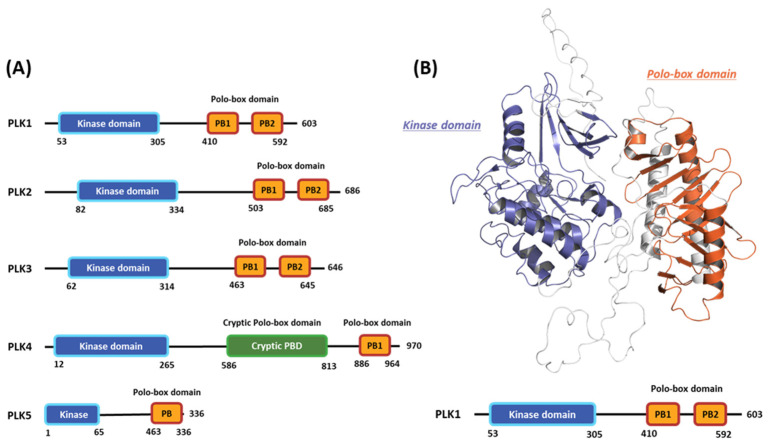
Domain information of the five human polo-like kinase (PLK) families and full-length structure of the PLK1 predicted by Alphafold2. (**A**) Numbers represent the amino acid sequence numbers. (**B**) Illustration of the kinase (blue) and polo-box (orange) domains.

**Figure 2 ijms-24-15639-f002:**
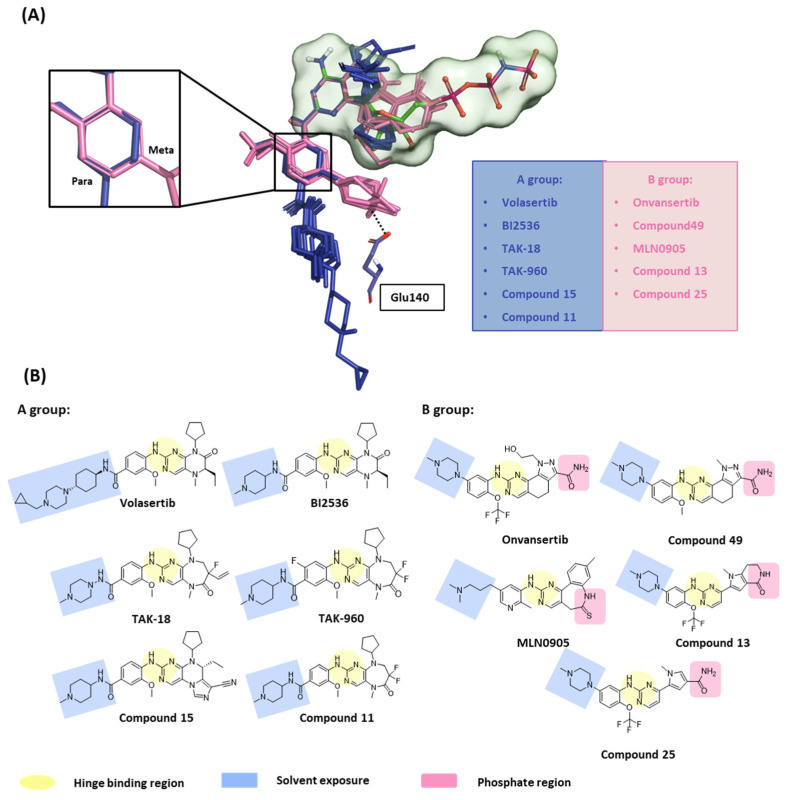
Structure-based selectivity pattern considerations. (**A**) Superposition of 11 crystal structures in the holo-form of human PLK1. Based on para and meta, PLK1 kinase domain inhibitors can be divided into two groups. The molecules of A group are represented in blue, and B in light pink. The inhibitors from the B group are PLK1 selective inhibitors toward PLK2 and 3. They all interact with Glu140 in PLK1. (**B**) Two-dimensional chemical structure of inhibitors from two groups. Hinge-binding regions are highlighted in light yellow, solvent exposure regions in light blue, and phosphate regions in light pink.

**Figure 3 ijms-24-15639-f003:**
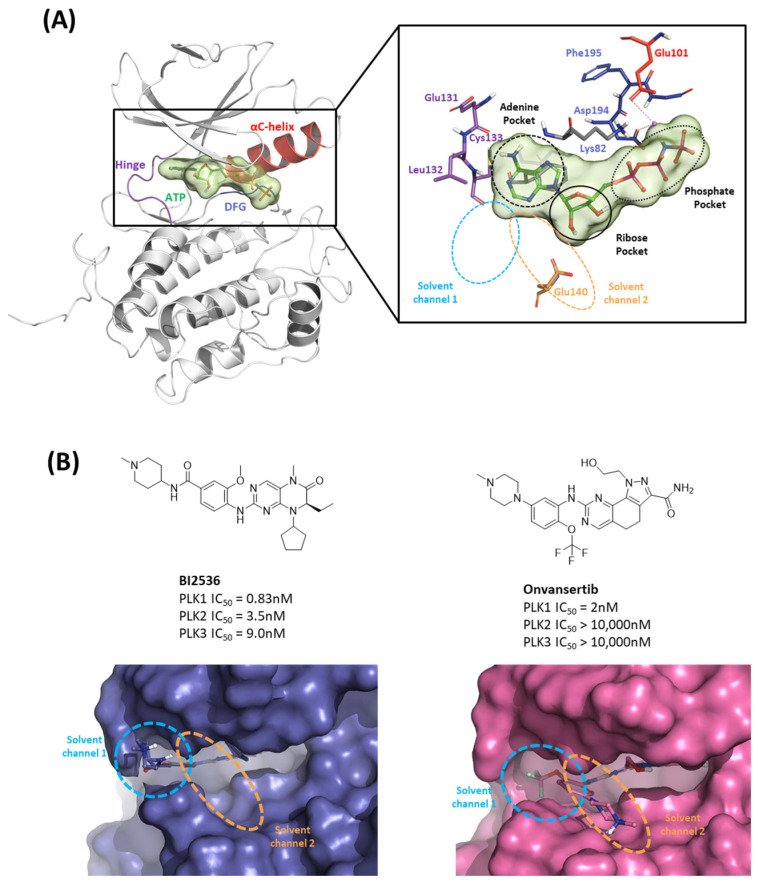
Characteristics of the ATP-binding pocket of the kinase domain in polo-like kinase 1 (PLK1). (**A**) Crystal structure of the human PLK1 kinase domain with ATP (PDB ID: 2OU7). The ATP-binding site is divided into adenine, ribose, and phosphate pockets and solvent Channels 1 and 2. (**B**) Comparison of the conformation change of the two crystal structures of the PLK1 complex with two inhibitors, BI2536 and Onvansertib. Nonselective inhibitor BI2536 only occupies solvent Channel 1, but potent PLK1 selective inhibitor Onvansertib binds solvent Channels 1 and 2.

**Figure 4 ijms-24-15639-f004:**
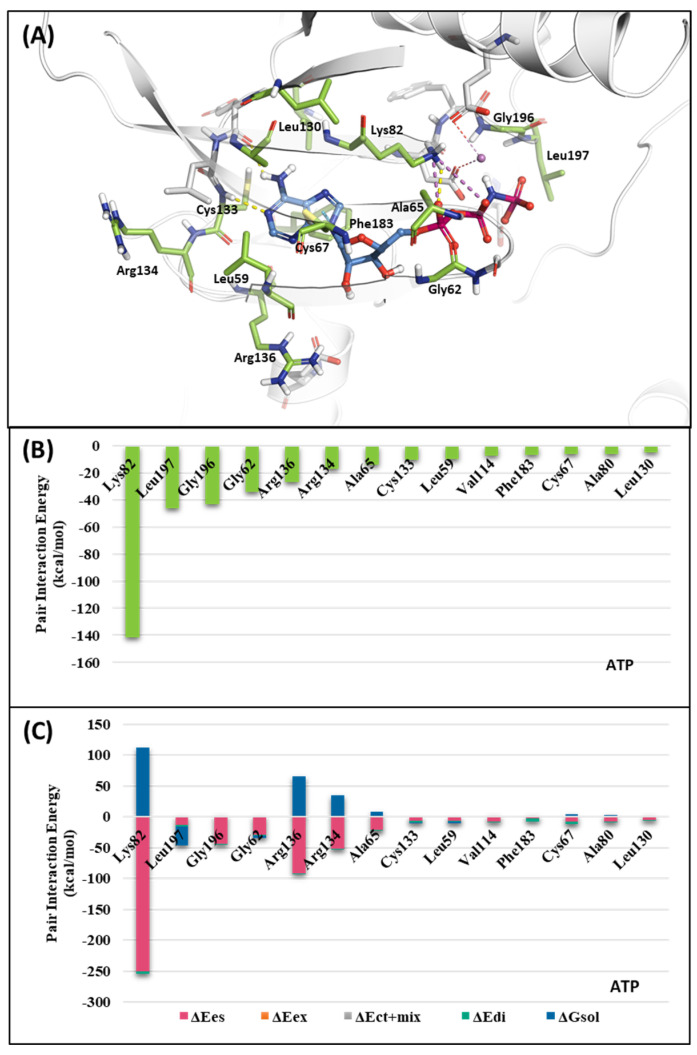
Fragment Molecular Orbital (FMO) analysis of ATP. (**A**) FMO results of the crystal structure of the PLK1 complex with the ATP analog. The ligand is blue; key residues are green. (**B**) PIE values of the significant residues in the ATP-binding site. (**C**) PIE decomposition analysis of these critical interactions.

**Figure 5 ijms-24-15639-f005:**
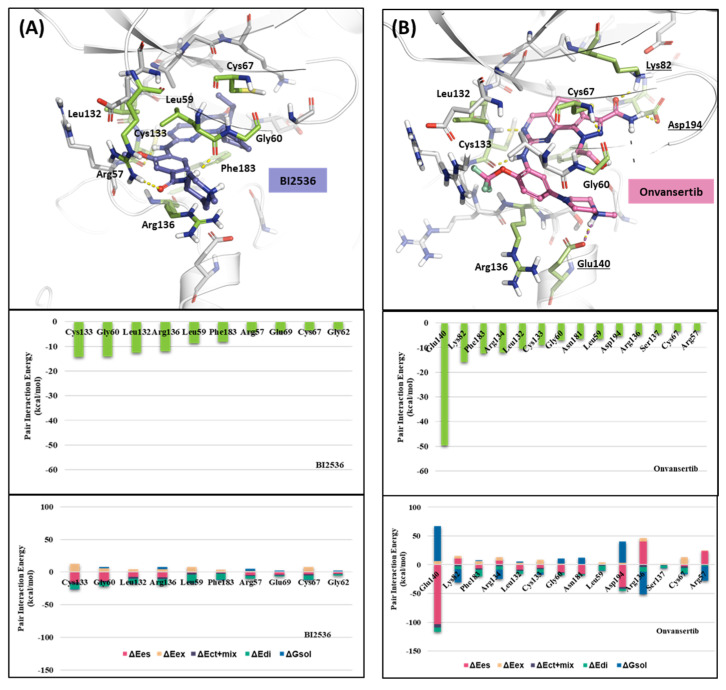
FMO analysis of BI2536 and Onvansertib. (**A**) FMO results of the crystal structure of the PLK1 complex with BI2536. The ligand is blue; the key residues are green. (**B**) FMO results of the crystal structure of the PLK1 complex with Onvansertib. The ligand is light pink, key residues are green, and nitrogen and oxygen atoms are blue and red. (**A**,**B**) Middle bar plots describe the PIE values of the significant residues in the ATP-binding site, whereas bottom bar plots describe the PIE decomposition analysis of these critical interactions.

**Figure 6 ijms-24-15639-f006:**
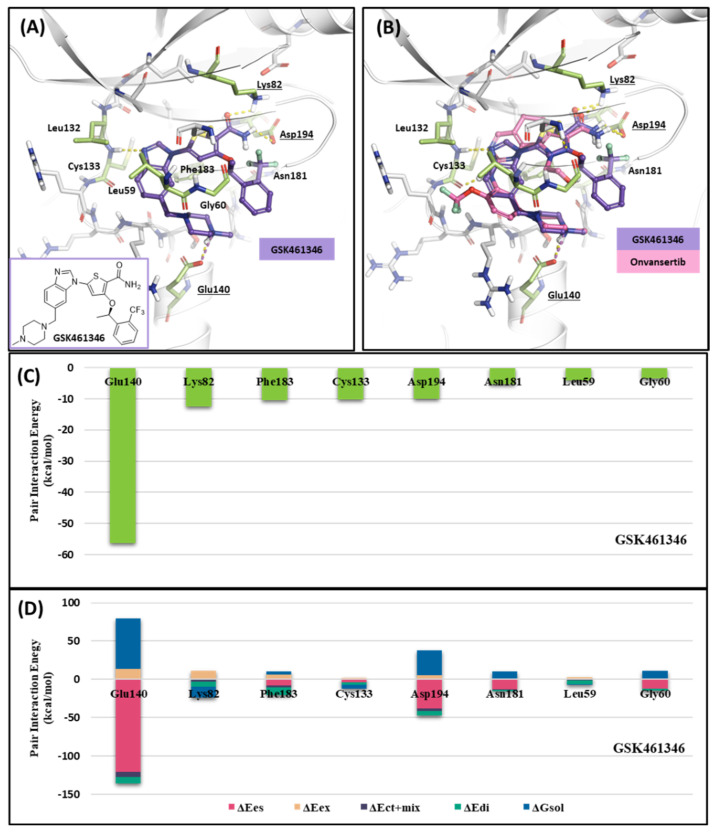
Docking structure and FMO analysis of GSK461364. (**A**) Docking structure of GSK461364. The ligand is purple; key residues are green. (**B**) Docking structure of GSK461364 overlayed with the crystal structure of Onvansertib (light pink). (**C**) Bar plot describing the PIE values of the significant residues in the ATP-binding site and GSK461364. (**D**) The bar plot describes the PIE decomposition analysis of these critical interactions.

**Figure 7 ijms-24-15639-f007:**
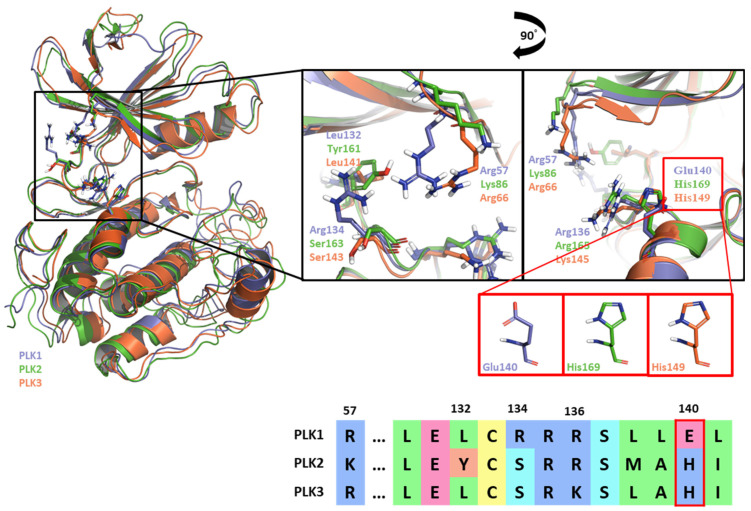
Superposition of the kinase domain of PLK1–3. Superposition of the crystal structures of the PLK1-3 and sequence alignment. Four residues occur around the ATP-binding site. Glu140 of PLK1 has a negative charge, except His169 and His149 of PLK2, and PLK3 have a positive charge.

**Figure 8 ijms-24-15639-f008:**
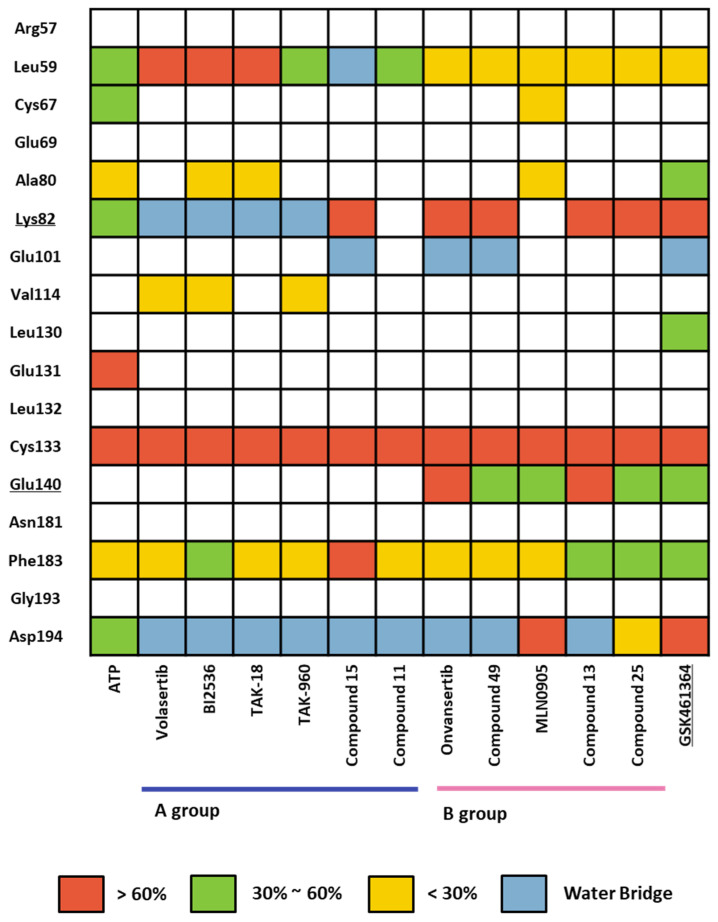
Heatmap of the protein–ligand contact from the molecular dynamic simulation of 13 ligands. Percentages represent interactions occurring during the simulation. Direct interaction <30% is yellow, 30% to 60% is green, and >60% is red. The water bridge is light blue.

**Figure 9 ijms-24-15639-f009:**
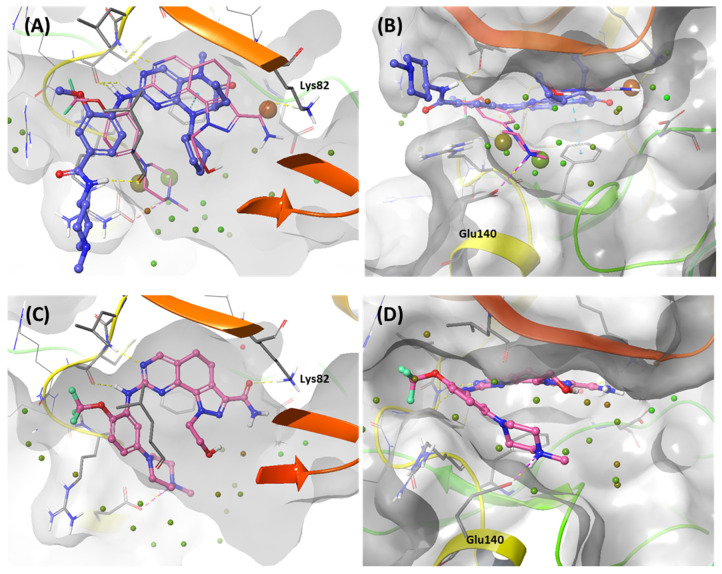
WaterMap analysis of the ATP-binding pocket of the kinase domain. Hydration sites are represented as spheres, with colors reflecting the predicted associated free energies. Green spheres signify favorable free energies, whereas red spheres indicate unfavorable free energies. (**A**,**B**) Onvansertib (light pink) is overlayed in the WaterMap of the crystal structure of BI2536, focusing on the hydration site of the phosphate-binding site and solvent Channel 2. (**C**,**D**) WaterMap of the crystal structure of Onvansertib.

**Figure 10 ijms-24-15639-f010:**
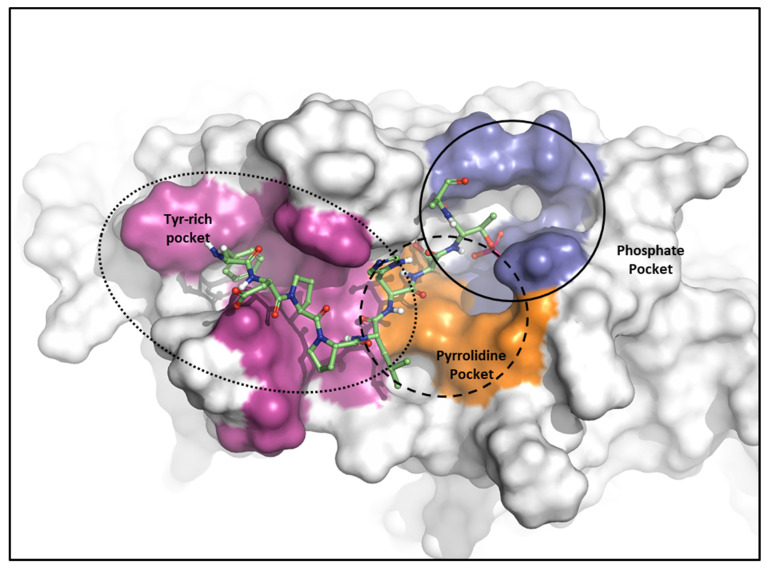
Characteristics of the PBD. Pockets of the PBD are divided into phosphate (blue), pyrrolidine (orange), and Tyr-rich (magenta) pockets.

**Figure 11 ijms-24-15639-f011:**
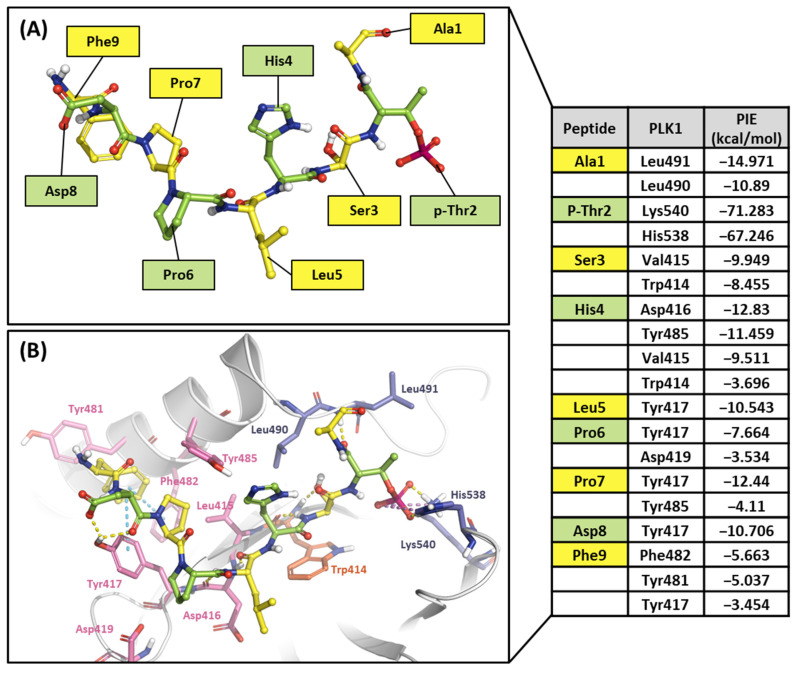
Hot-spot analysis of substrate peptide complex with the PBD of PLK1. (**A**) The substrate peptide was divided into nine fragments: Ala1, p-Thr2, Ser3, His4, Leu5, Pro6, Pro7, Asp8, and Phe9 (green and yellow sticks). (**B**) PLK1 PBD hot-spot residues are blue, orange, and light pink sticks. PIE values are described in the right table.

**Figure 12 ijms-24-15639-f012:**
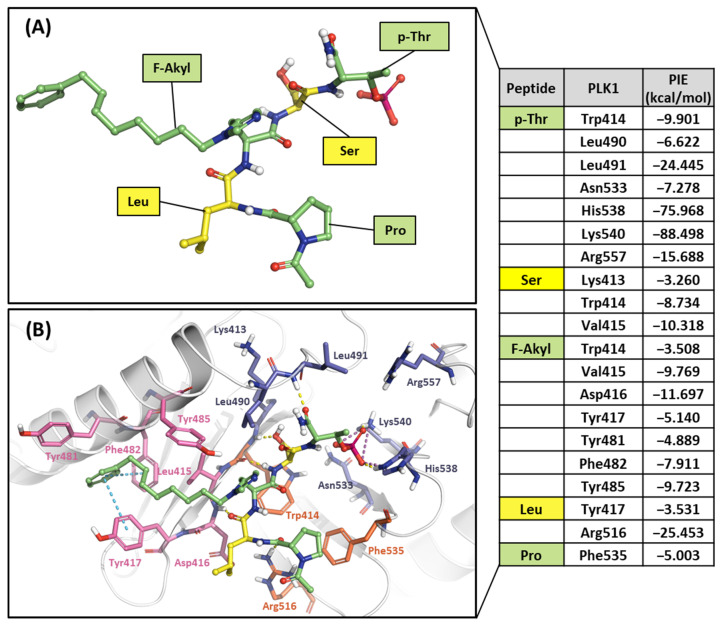
Hot-spot analysis of the 4j complex with the PBD of PLK1. (**A**) Fragments of 4j: p-Thr, Ser, F-Akyl, Leu, and Pro (green and yellow sticks). (**B**) Significant interaction residues in the PBD pocket are blue, orange, and light pink sticks. PIE values are described in the right table.

**Figure 13 ijms-24-15639-f013:**
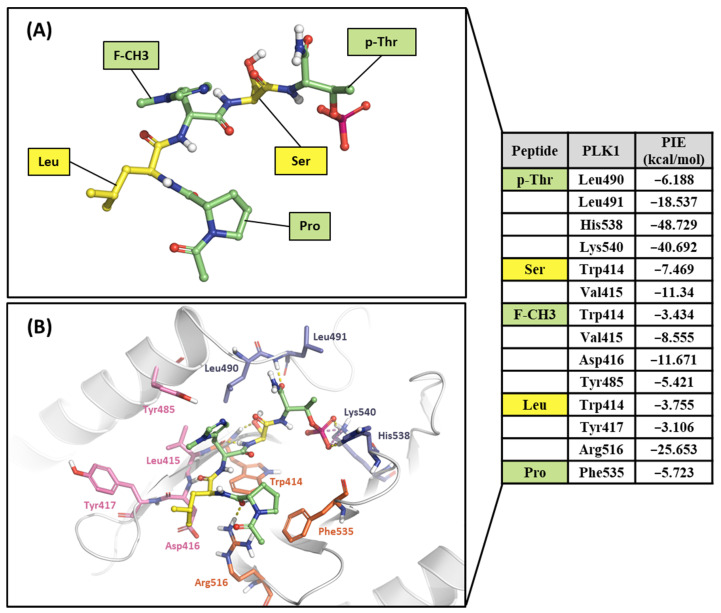
Hot-spot analysis of the 4a complex with the PBD of PLK1. (**A**) Fragments of 4a: p-Thr, Ser, F-CH3, Leu, and Pro (green and yellow sticks). (**B**) Significant interaction residues in the PBD pocket are blue, orange, and light pink sticks. PIE values are described in the right table.

**Figure 14 ijms-24-15639-f014:**
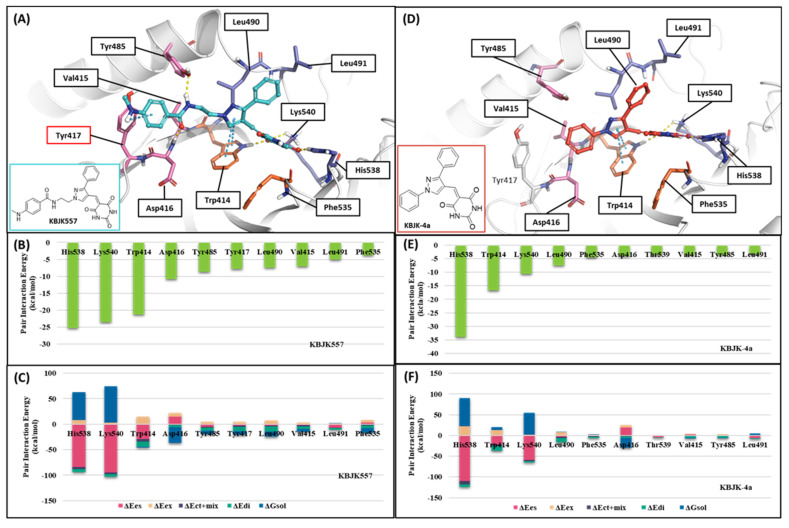
FMO analysis of KBJK557 and KBJK-4a. (**A**) FMO results of the structure from Frame 489 of the molecular dynamic simulation. The ligand is blue; key protein residues are blue, orange, and light pink sticks. (**B**) Bar plot describing the PIE values of the significant residues in the PBD. (**C**) Bar plot describing the PIE decomposition analysis of these critical interactions. (**D**) FMO results of the docking structure of the KBJK-4a. The ligand is blue; key protein residues are blue, orange, and light pink sticks. (**E**) Bar plot describing the PIE values of the significant residues in the PBD. (**F**) Bar plot describing the PIEDA of these critical interactions.

**Figure 15 ijms-24-15639-f015:**
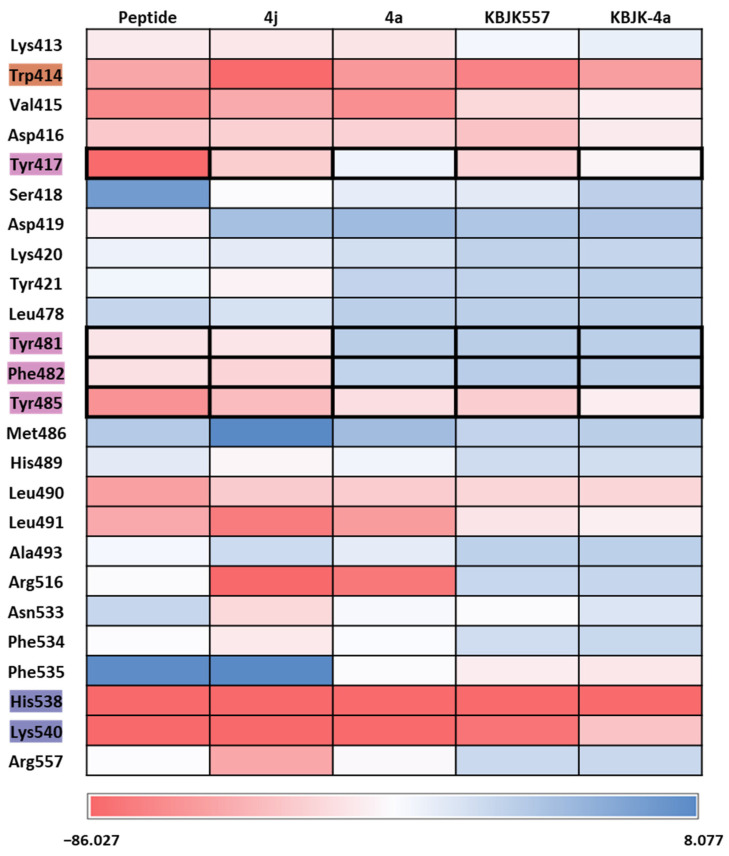
Heatmap of the FMO results of five structures. The FMO results of the complex structure of the PBD of PLK1 complex with substrate peptide, 4j and 4a, KBJK557, and KBJK-4a. The critical residues are highlighted blue for the phosphate pocket, orange for the pyrrolidine pocket, and pink for the Tyr-rich pocket. The pair interaction energy was summed for each residue. Darker red indicates a lower energy value.

**Figure 16 ijms-24-15639-f016:**
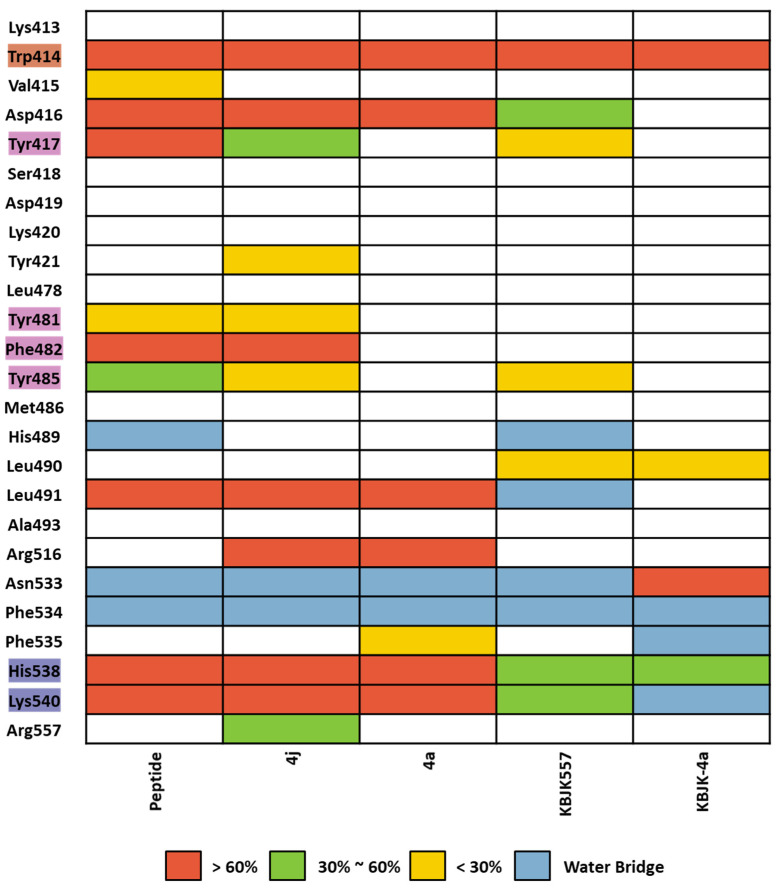
Heatmap of the protein–ligand contact from the molecular dynamic simulation of 5 ligands. Percentages represent interactions occurring during the simulation. The critical residues are highlighted blue for the phosphate pocket, orange for the pyrrolidine pocket, and pink for the Tyr-rich pocket. Direct interaction <30% is yellow, 30% to 60% is green, and >60% is red. The water bridge is light blue.

**Figure 17 ijms-24-15639-f017:**
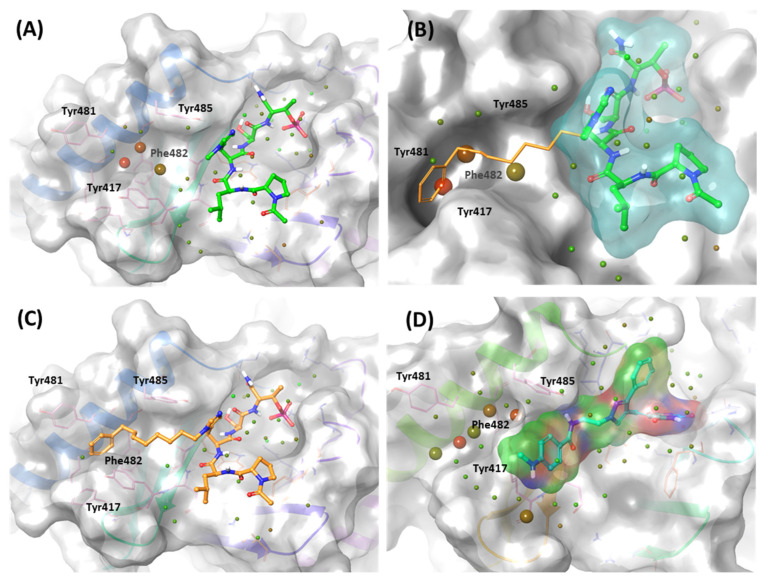
WaterMap analysis of the binding pocket of the PBD. Hydration sites are represented as spheres, with colors reflecting their predicted free energies. Green spheres signify favorable free energies; red spheres indicate unfavorable free energies. (**A**) WaterMap analysis of the 4a (lime) binding structure in the PBD. (**B**) Focusing on the Tyr-rich pocket in the 4a (cyan surface) water map analysis with 4j superposition (orange stick). (**C**) WaterMap analysis of the crystal structure of the 4j binding in the PBD. (**D**) WaterMap focusing on KBJK557 (surfaced with a partial charge) and hydration sites in the Tyr-rich pocket.

**Table 1 ijms-24-15639-t001:** PLK1 kinase domain inhibitors in clinical trials.

Drugs	Potency	Selectivity to PLK2 and PLK3	Clinical Phase
BI2536	PLK1 IC_50_ = 0.83 nM	PLK2 IC50 = 3.5 nM	II
		PLK3 IC_50_ = 9.0 nM	
Volasertib	PLK1 IC_50_ = 0.87 nM	PLK2 IC_50_ = 5 nM	I/II/III
		PLK3 IC_50_ = 56 nM	
Onvansertib	PLK1 IC_50_ = 2 nM	PLK2 IC_50_ > 10,000 nM	II/III
		PLK3 IC_50_ > 10,000 nM	
TAK-960	PLK1 IC_50_ = 0.8 nM	PLK2 IC_50_ = 16.9 nM	I
		PLK3 IC_50_ = 50.2 nM	
GSK461364	PLK1 Kiapp < 0.5 nM	PLK2 Kiapp = 860 nM	I
		PLK3 Kiapp = 1000 nM	

## Data Availability

Data is contained within the article or Appendix A.

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
