# Peer review of "Leveraging the Fragment Molecular Orbital Method to Explore the PLK1 Kinase Binding Site and Polo-Box Domain for Potent Small-Molecule Drug Design"

_ijms, 2023, doi:10.3390/ijms242115639_

Round 1
Reviewer 1 Report
Comments and Suggestions for Authors
The Authors have done an in-depth computational analysis of existing PLK1 inhibitors, both those targeting the ATP binding pocket and the newer molecules which target the Polo box domain (PBD).
Unfortunately the manuscript is not well written, it is repetitious with a whole paragraph in the introduction actually repeated and sentences in the abstract that don't make sense.
The authors also need to be more transparent about which experimental structural models specifically they have used to base their computational analysis on in the main text citing the specific PDB codes appropriately rather than just having a table in the supplementary file.
The conclusions drawn about the ATP competitive molecules are not surprising based on the wealth of research that has been carried out on ATP competitive inhibitors binding to kinase domains. It would have been nice if the authors had actually compared their modelling data to the experimental work carried out by others. This would validate their approach and give greater weight to their findings on the PBD binding molecules.
Overall, this is a solid study however the manuscript needs significant editing before it is suitable for publication.
Comments on the Quality of English LanguageThe quality of the english is fine, however the manuscript as a whole is not well written, with typos and repeated paragraphs. It is currently not engaging to read and does not convey the research as well as it could.
Author Response
Dear Reviewer,
I hope this letter finds you well. I am writing to express our gratitude for taking the time to review our paper. Your feedback is incredibly valuable to us, and we sincerely appreciate your thoughtful comments. We have carefully considered your suggestions and have made revisions accordingly.
- Comment: “It is repetitious with a whole paragraph in the introduction actually repeated and sentences in the abstract that don't make sense”
Revision: According to your first comment, we removed the repeated sentences in the paper. - Comment: “The authors also need to be more transparent about which experimental structural models specifically they have used to base their computational analysis on in the main text citing the specific PDB codes appropriately rather than just having a table in the supplementary file.”
Revision: Following the second comment, we add the specific PDB codes to the specific structures. - Comment: “The conclusions drawn about the ATP competitive molecules are not surprising based on the wealth of research that has been carried out on ATP competitive inhibitors binding to kinase domains. It would have been nice if the authors had actually compared their modeling data to the experimental work carried out by others. This would validate their approach and give greater weight to their findings on the PBD binding molecules.”
Revision: Following the last comment about comparing our modeling data to the experimental work from other groups, we mentioned in the results part many times. In addition, we also add the sentence “Our findings obtained through computation are in agreement with the results of biological assays conducted by other research groups.” in the conclusion part.
Please see the details of the revision of the paper in the attachment file.
Your expertise and insights have greatly contributed to the improvement of our work, and we are truly thankful for your support. If you have any further suggestions or feedback, please do not hesitate to let us know.
Once again, thank you for your time and valuable input. We are sincerely grateful for your help.
Warm regards,
Haiyan Jin

Reviewer 2 Report
Comments and Suggestions for Authors
The authors have effectively communicated their research, which employs the Fragment Molecular Orbital (FMO) method to develop small-molecule inhibitors for PLK1. Their study also underscores the significance of identifying "hot spot" residues crucial for achieving selective and robust binding to PLK1. These findings hold substantial promise for the rational design of novel PLK1 inhibitors, potentially advancing the field of anticancer therapeutics.
Overall, the manuscript is well-prepared and appears to be in a suitable state for publication.
Author Response
Dear Reviewer,
I hope this letter finds you well. I am writing to express our gratitude for taking the time to review our paper. Your feedback is incredibly valuable to us, and we sincerely appreciate your thoughtful comments.
Please see the details of the revision of the paper in the attachment file.
Your expertise and insights have greatly contributed to the improvement of our work, and we are truly thankful for your support. If you have any further suggestions or feedback, please do not hesitate to let us know.
Once again, thank you for your time and valuable input. We are sincerely grateful for your help.
Warm regards,
Haiyan Jin

Reviewer 3 Report
Comments and Suggestions for Authors
The manuscript is quite well written. The topic is interesting. I have some suggestions:
1) Abstract. This comparison enhanced our understanding of the structure-activity relationships of these inhibitors, underscoring the efficacy of the FMO method in identifying critical binding features and predicting binding modes for small-molecule ligands. Furthermore, Our research spotlighted "hot spot" residues instrumental for selective and robust PLK1 binding. Our findings offer profound insights, priming the rational design of innovative potential PLK1 inhibitors with significant implications for developing anticancer therapeutics.
Please, improve the description of study results to increase the citation index.
2) 1. Introduction: Polo-like kinases (PLKs) form a family of serine/tyrosine kinase proteins with wide distribution in eukaryotic cells and play crucial roles in various cell-cycle phases. Currently, the PLK protein family comprises five members: PLK1, PLK2, PLK3, PLK4, and PLK5 (Figure 1(A)). Among these members, PLK1 has undergone the most comprehensive research to understand the regulatory mechanisms influencing its functions and potential as a target for drug design [1]. Please, support all these sentences with the correct references.
3) In this study, we employed the FMO/3D-SPIE analysis to investigate PPIs between the substrate peptide and the PBD of PLK1 at the quantum mechanical level. We utilized the FMO method to analyze interactions within the ATP-binding site, as well as with PBDs, as part of our PLK1 hot-spot analysis. This research investigates the ATP and substrate peptide binding sites of the KD and PBD of PLK1, respectively. We explored the interactions of inhibitors BI2536, Onvansertib, and GSK461364, shedding light on the key residues responsible for selectivity. Our FMO analysis furnished insights into their respective binding modes. Furthermore, our study emphasized the significance of particular pockets in PBD binding. Molecular dynamic (MD) simulations and solvation analysis further substantiated our findings, pinpointing potential avenues for enhancing inhibitor efficacy. Taken together, this research provides invaluable insights for drug design and deepens our understanding of the binding mechanisms of PLK1. Please report all the objectives of the work in a more schematic way to clarify them and increase the citation index of the article.
4) 2. Result: 2.1. Hot-spot Analysis for the Kinase Domain in PLK1. Please underline in section the most important results of the study to clarify them.
5) 3. Discussion: A group of PLKs, consisting of five serine/threonine kinases, can be found in different eukaryotic organisms[59]. These kinases play a crucial role in regulating cell proliferation, particularly in controlling the progression of the cell cycle [2]. The PLK1 protein consists of an N-terminal serine/threonine kinase domain (KD) and a C-terminal repeat of the Polo-box domain (PBD), with the phosphorylation of the latter directly affecting the enzymatic activity of PLK1 [53]. The PBD recognizes phosphorylated serine/threonine protein substrates to regulate PLK KD's phosphorylation activity [20]. Each domain comprises druggable binding sites, including an ATP-binding site in the KD and a substrate peptide binding site in the PBD.
Please, summarise here the most important results of the study.
6) 4. Conclusion In conclusion, our study integrated the FMO methodology, MD simulations, and solvation analyses to investigate the KD and PBD domains of the PLK1 protein. The insights shed light on the intricate interactions and binding preferences of various PLK1 inhibitors. Our findings underscore the consistent binding patterns of ATP-competitive inhibitors to PLK1, emphasizing their selectivity. Furthermore, our investigation into the PBD's substrate peptide binding site pinpointed crucial residues vital for interactions, paving the way to bolster inhibitor potency. Collectively, these revelations considerably elevate our comprehension of the mechanisms dictating binding, selectivity, and the prospective therapeutic avenues for PLK1-targeting inhibitors. Grounded in our research, we are poised to contribute invaluable perspectives for the next generation of bivalent PLK1 inhibitor designs. This wealth of knowledge will indubitably influence the blueprint of upcoming pharmaceuticals, aiming to craft more efficacious and discerning inhibitors that modulate cell growth and orchestrate the cell cycle. Please, underline the novelty of the study and the possible clinical implicantions of the study.
Comments on the Quality of English LanguageMinor changes of English language are required
Author Response
Dear Reviewer,
I hope this letter finds you well. I am writing to express our gratitude for taking the time to review our paper. Your feedback is incredibly valuable to us, and we sincerely appreciate your thoughtful comments. We have carefully considered your suggestions and have made revisions accordingly.
- Comment: “Abstract. This comparison enhanced our understanding of the structure-activity relationships of these inhibitors, underscoring the efficacy of the FMO method in identifying critical binding features and predicting binding modes for small-molecule ligands. Furthermore, Our research spotlighted "hot spot" residues instrumental for selective and robust PLK1 binding. Our findings offer profound insights, priming the rational design of innovative potential PLK1 inhibitors with significant implications for developing anticancer therapeutics.
Please, improve the description of study results to increase the citation index.”
Revision: According to your first comment, we revised this part to “Our investigation further entailed a comparative analysis of various PLK1 inhibitors, each characterized by distinct structural attributes, helping us gain a better understanding of the relationship between molecular structure and biological activity. The FMO method was particularly effective in identifying key binding features and predicting binding modes for small-molecule ligands. Our research also highlighted specific "hot spot" residues that played a critical role in the selective and robust binding of PLK1. These findings provide valuable insights that can be used to design new and effective PLK1 inhibitors, which can have significant implications for developing anticancer therapeutics.” - Comment: “Introduction: Polo-like kinases (PLKs) form a family of serine/tyrosine kinase proteins with wide distribution in eukaryotic cells and play crucial roles in various cell-cycle phases. Currently, the PLK protein family comprises five members: PLK1, PLK2, PLK3, PLK4, and PLK5 (Figure 1(A)). Among these members, PLK1 has undergone the most comprehensive research to understand the regulatory mechanisms influencing its functions and potential as a target for drug design [1].
Please, support all these sentences with the correct references.”
Revision: According to your second comment, we add two references to the first and second sentences. “Polo-like kinases (PLKs) form a family of serine/tyrosine kinase proteins with wide distribution in eukaryotic cells and play crucial roles in various cell-cycle phases [1]. Currently, the PLK protein family comprises five members: PLK1, PLK2, PLK3, PLK4, and PLK5 (Figure 1(A)) [2]. Among these members, PLK1 has undergone the most comprehensive research to understand the regulatory mechanisms influencing its functions and potential as a target for drug design [3].” - Comment: “In this study, we employed the FMO/3D-SPIE analysis to investigate PPIs between the substrate peptide and the PBD of PLK1 at the quantum mechanical level. We utilized the FMO method to analyze interactions within the ATP-binding site, as well as with PBDs, as part of our PLK1 hot-spot analysis. This research investigates the ATP and substrate peptide binding sites of the KD and PBD of PLK1, respectively. We explored the interactions of inhibitors BI2536, Onvansertib, and GSK461364, shedding light on the key residues responsible for selectivity. Our FMO analysis furnished insights into their respective binding modes. Furthermore, our study emphasized the significance of particular pockets in PBD binding. Molecular dynamic (MD) simulations and solvation analysis further substantiated our findings, pinpointing potential avenues for enhancing inhibitor efficiency. Taken together, this research provides invaluable insights for drug design and deepens our understanding of the binding mechanisms of PLK1.
Please report all the objectives of the work in a more schematic way to clarify them and increase the citation index of the article.”
Revision: Following your comment, we revised this part to “In this study, we employed Fragment Molecular Orbital (FMO) and 3D-SPIE analyses to investigate protein-protein interactions (PPIs) between the substrate peptide and the Po-lo-Box Domain (PBD) of Polo-like kinase 1 (PLK1) at the quantum mechanical level. We uti-lized the FMO method for an in-depth analysis of interactions within the ATP-binding site and the PBDs, integral to our hot-spot analysis of PLK1. This research delves into the ATP and substrate peptide binding sites of the Kinase Domain (KD) and PBD of PLK1, respec-tively. We examined the interactions with inhibitors BI2536, Onvansertib, and GSK461364, illuminating the key residues responsible for selectivity. Our FMO analysis provided in-sights into their distinct binding modes, emphasizing the significant role of specific PBD binding pockets. We explored PBD's binding to peptides and small molecules, pinpointing the "hot spot" regions crucial for potency. Molecular dynamics (MD) simulations and solvation analysis further validated our results, highlighting potential strategies for aug-menting inhibitor efficacy. Collectively, this research offers invaluable insights into drug design, enhancing our understanding of PLK1's binding mechanisms.” - Comment: “Result: 2.1. Hot-spot Analysis for the Kinase Domain in PLK1.
Please underline in section the most important results of the study to clarify them.”
Revision: Following your comment, we underlined “result 2.1.Hot-spot Analysis for the Kinase Domain in PLK1” and “result 2.2.Hot-spot Analysis for the Polo-Box Domain in PLK1”. - Comment: “Discussion: A group of PLKs, consisting of five serine/threonine kinases, can be found in different eukaryotic organisms[59]. These kinases play a crucial role in regulating cell proliferation, particularly in controlling the progression of the cell cycle [2]. The PLK1 protein consists of an N-terminal serine/threonine kinase domain (KD) and a C-terminal repeat of the Polo-box domain (PBD), with the phosphorylation of the latter directly affecting the enzymatic activity of PLK1 [53]. The PBD recognizes phosphorylated serine/threonine protein substrates to regulate PLK KD's phosphorylation activity [20]. Each domain comprises druggable binding sites, including an ATP-binding site in the KD and a substrate peptide binding site in the PBD.
Please, summarise here the most important results of the study.”
Revision: According to your comment, we added the summary sentence following this part: “Our computational study investigates two druggable binding sites to improve selectivity in the KD and potency in the PBD.” - Comment: “Conclusion In conclusion, our study integrated the FMO methodology, MD simulations, and solvation analyses to investigate the KD and PBD domains of the PLK1 protein. The insights shed light on the intricate interactions and binding preferences of various PLK1 inhibitors. Our findings underscore the consistent binding patterns of ATP-competitive inhibitors to PLK1, emphasizing their selectivity. Furthermore, our investigation into the PBD's substrate peptide binding site pinpointed crucial residues vital for interactions, paving the way to bolster inhibitor potency. Collectively, these revelations considerably elevate our comprehension of the mechanisms dictating binding, selectivity, and the prospective therapeutic avenues for PLK1-targeting inhibitors. Grounded in our research, we are poised to contribute invaluable perspectives for the next generation of bivalent PLK1 inhibitor designs. This wealth of knowledge will indubitably influence the blueprint of upcoming pharmaceuticals, aiming to craft more efficacious and discerning inhibitors that modulate cell growth and orchestrate the cell cycle.
Please, underline the novelty of the study and the possible clinical implications of the study”
Revision: Following your comment, we revised this part to “In conclusion, our study has integrated FMO methodology, MD simulations, and solvation analyses to delve into the KD and PBD domains of the PLK1 protein. Our find-ings obtained through computation are in agreement with the results of biological assays conducted by other research groups. These insights illuminate the complex interactions and binding preferences among various PLK1 inhibitors. Importantly, our research highlights the consistent binding patterns of ATP-competitive inhibitors to PLK1, underscoring their selectivity. Additionally, our exploration of the PBD's substrate peptide binding site identified crucial residues essential for interactions, suggesting pathways to enhance inhibitor efficacy. Collectively, these discoveries significantly enhance our understanding of the mechanisms governing binding, selectivity, and potential therapeutic pathways for PLK1-targeting inhibitors. Drawing from our research, we can provide valuable guidance for designing future bivalent PLK1 inhibitors with meaningful clinical implications. This wealth of information will undoubtedly shape the development strategies of forthcoming pharmaceuticals, guiding the creation of more effective and selective inhibitors that control cell growth and direct the cell cycle.”
Please see the details of the revision of the paper in the attachment file.
Your expertise and insights have greatly contributed to the improvement of our work, and we are truly thankful for your support. If you have any further suggestions or feedback, please do not hesitate to let us know.
Once again, thank you for your time and valuable input. We are sincerely grateful for your help.
Warm regards,
Haiyan Jin

Round 2
Reviewer 1 Report
Comments and Suggestions for Authors
The changes the authors have made have improved this manuscript.
Reviewer 3 Report
Comments and Suggestions for Authors
No further comments
Comments on the Quality of English LanguageMinor changes are required